# Safety Profile and Issues of Subcutaneous Immunotherapy in the Treatment of Children with Allergic Rhinitis

**DOI:** 10.3390/cells11091584

**Published:** 2022-05-09

**Authors:** Anang Endaryanto, Ricardo Adrian Nugraha

**Affiliations:** 1Division of Pediatric Allergy and Immunology, Department of Child Health, Dr. Soetomo General Academic Hospital, Faculty of Medicine, Universitas Airlangga, Surabaya 60285, Indonesia; 2Department of Cardiology and Vascular Medicine, Dr. Soetomo General Academic Hospital, Faculty of Medicine, Universitas Airlangga, Surabaya 60285, Indonesia; ricardo.adrian.nugraha-2019@fk.unair.ac.id

**Keywords:** allergic rhinitis, house dust mites, subcutaneous immunotherapy, safety, efficacy

## Abstract

This study aims to evaluate safety issues of house dust mite subcutaneous immunotherapy (SCIT) among allergic rhinitis (AR) children. A retrospective cohort study was done between 2015 and 2020 to investigate the side effects of SCIT among AR children caused by a house dust mite allergy. Among 1098 patients who received house dust mite subcutaneous immunotherapy injections, 284 patients (25.87%) had side effects (SE). SE were found to be 699 times higher or in 2.27% of the 30,744 subcutaneous immunotherapy injections. A total of 17.9% of the patients had local SE during SCIT administration. Systemic side effects occurred in 8.38% of children receiving SCIT and in 0.53% of the total population who received SCIT injections. Only 2/92 (2.18%) of patients suffered an allergic reaction within 30 minutes of injection and these patients responded well to antiallergic medication. Severe anaphylaxis occurred in 0.091% of the 1098 patients in the SCIT group and in 0.0033% of the 30,774 SCIT injections. Systemic SE after SCIT occurred in 8.38% of patients receiving SCIT or 0.53% of the total number of SCIT injections. Anaphylactic episodes occurred in 16 patients (1.46%) and 15 patients (1.37%) who had first and second episodes. One severe attack was found and it was resolved with adrenaline. This study demonstrates that in pediatric patients with AR who received HDM SCIT for 18 months with high adherence, some experienced significant local SE and systemic SE caused by SCIT, but this did not interfere with the course of AR treatment or the effectiveness of SCIT.

## 1. Introduction

Allergen immunotherapy (AIT), both subcutaneous immunotherapy (SCIT) and sublingual immunotherapy (SLIT), is adequate to relieve symptoms and medication use in subjects with allergic rhinitis (AR) with or without allergic asthma [1,2,3,4]. The AIT commonly used in Indonesia is SCIT. For Indonesia’s private healthcare system, only one private allergy clinic that provides special services for allergic children is equipped with SCIT services. It receives special referrals for allergic children who need SCIT from general practitioners, pediatricians, and other specialists in all regions of Indonesia. SCIT is considered safe and well-tolerated when injected with good medical regulation by trained personnel who can recognize and treat systemic reactions early [5]. SCIT in children has always been a dilemma for doctors. On the one hand, there is strong evidence supporting efficacy [6,7]; on the other hand, children always show more robust resistance to SCIT injection. In addition, they often have upper respiratory tract infections that mimic allergy symptoms that can occur after SCIT and can be considered a side effect (SE) of SCIT.

Many studies have linked nonadherence to a schedule and discontinuation of SCIT with systemic and local side effects of SCIT [8,9,10,11,12,13]. Still, the systemic and local side-effect profiles of SCIT in these studies were not compared with those who did not receive SCIT (control), so it can be concluded that the reported side effects occur by chance (co-incidental) or because of the impact of the course of the allergic disease (including AR), or because of the effects of standard drugs (other than SCIT) administered or other diseases. In addition, not many studies have reported the profiles of systemic and local side effects in patients who are relatively adherent to the SCIT schedule and can complete SCIT to the maintenance phase. Many studies collect data from various SCIT administration sites. The location of SCIT administration seems to affect the safety profile, where the safety profile is better when in service locations with better medical surveillance facilities [10]. Patient compliance with the SCIT schedule and patient satisfaction also depend on the professionalism of the SCIT administration site.

The essential requirement of our study was to obtain estimates of the incidence of side effects (SE) purely from SCIT. Adverse events (AE) following SCIT are often reported as SCIT side effects (SE), even though they could be coincidences that do not originate from SCIT [8,9,10,11,12,13]. Urticaria, angioedema, asthma, and rhinoconjunctivitis occurring after SCIT may also occur in children with allergic rhinitis who do not receive SCIT. By comparing these symptoms in the SCIT group with the non-SCIT group in a large population, we can estimate how significant the side effects are that are purely caused by SCIT.

### Objective

In this observational study, we collected data regarding side effects of SCIT from allergic rhinitis children caused by a house dust mite (HDM) allergy, with/without asthma, who received SCIT with high adherence rates over an 18-month SCIT duration, compared with data on adverse event reactions (AE) that are experienced by non-SCIT patients as controls, in private practice facilities.

## 2. Materials and Methods

### 2.1. Ethics Approval

This study was conducted in strict accordance with the Declaration of Helsinki guidelines and approved for exempt review by the Research Ethics Committee of the Dr. Soetomo General Academic Hospital (protocol code 0389/105/XI/I/2020 version 4 and date of approval 25 January 2021). The Health Research Ethics Committee of the Dr. Soetomo General Academic Hospital stated that the research protocol document above complies with the Office for Human Research Protections (OHRP) under the U.S. Department of Health and Human Services (HHS) Regulation 45 CFR section 46 for exempt review. General data—including name, address, age, gender, weight, height, and telephone number—and specific data regarding side effects (SE) and adverse events (AE) were collected and recorded for all subjects. Likewise, certain data, e.g., allergens, asthma comorbidities, their duration, and details on other allergies and their medications, were also recorded. Written informed consent was waived because this study was limited to using data from existing medical records. When starting therapy, the subjects and their parents were instructed that their medical record data would be used for observational research. Implied consent was obtained from parents or caregivers.

### 2.2. Study Design

A retrospective cohort study was done from 2015 until 2020 to investigate any side effects (SE) of SCIT among rhinitis children caused by a house dust mite (HDM) allergy, with/without asthma, who received SCIT with high adherence rates within 18 months of SCIT duration, compared with data on adverse reactions (AR) experienced by non-SCIT patients as controls, in private practice facilities. The SCIT and non-SCIT groups consisted of 1098 subjects each (Figure 1). The SCIT group received standard therapy plus SCIT HDM, whereas the non-SCIT group only received standard treatment. This study carried on data from a private allergy clinic of a pediatric allergy consultant in Surabaya, East Java, Indonesia, from 1 January 2015 to 31 March 2020.

The allergic condition was defined by a typical clinical history, positive skin prick tests, and serum specific IgE for HDM allergens. The diagnosis of AR and asthma was consistent with ARIA [14] and GINA [15]. The exclusion criteria for subjects included subjects with an abnormal shape in the anatomy of the nose and paranasal sinuses, and patients diagnosed with cancer, autoimmune diseases, cerebral palsy, and Down syndrome.

### 2.3. Materials

House dust mite allergen immunotherapy (Teaching Industry Allergen by Airlangga University—Dr. Soetomo General Academic Hospital, Surabaya, Indonesia) used was *Dermatophagoides pteronyssinus* extract with 11.3–26.6 ng/mL via subcutaneous injection [16,17,18]. Immunotherapy is provided in two phases: the build-up and maintenance phases. The build-up phase consists of injections given weekly; in the maintenance phase, injections are offered every three weeks. The dose of immunotherapy used every week varies from 0.1 cc (first week) to 0.15 cc (second week), 0.22 cc (third week), 0.32 cc (fourth week), 0.48 cc (fifth week), 0.72 cc (sixth week), 1 cc (seventh week), 0.1 cc (eighth week), 0.15 cc (ninth week), 0.22 cc (tenth week), 0.32 cc (eleventh week), 0.48 cc (twelfth week), 0.72 ccs (thirteenth week), and 1 cc (fourteenth week), and the following week [17]. The type of HDM allergen content in SCIT used in this study was based on previous research in Indonesia, which stated that the most common types of HDM allergen found in Indonesia were *Dermatophagoides pteronyssinus* (87%), *Dermatophagoides farinae* (7%), and *Bromia tropicalis* (6%) [19]. Another study on HDM in Indonesia informed us that *Bromia tropicalis* is the least common compared to *Dermatophagoides pteronyssinus* and *Dermatophagoides farinae*. *Dermatophagoides pteronyssinus* can be found in various places such as beds, floors, and sofas. Meanwhile, *Dermatophagoides farinae* is often found in sofas [20].

Information was collected from the SCIT administration (including manufacturer, dose, date of injection, the concentration of extract, the dose administered, and physician explanation, such as insertion of the injection site, type, and severity of reaction) prospectively filled out by physicians at SCIT administration sessions. All of the subjects we studied were from a large cohort of public and private pediatric immunological allergy patients managed by the Division of Allergy-Immunology, Department of Pediatrics, Faculty of Medicine, Universitas Airlangga, Dr. Soetomo General Academic Hospital for research and development of allergy care in children since 2001. All parents of patients in the cohort have agreed that their children will receive medical treatment, monitor the course of the disease, and disease outcomes by standard operating procedures for research and patient care in the Allergy-Immunology Division, Department of Pediatrics, Faculty of Medicine, Airlangga University, Dr. Soetomo General Academic Hospital. A monitoring form for allergy patients who received SCIT or not (which was held by the doctor and by the patient’s parent/caregiver) was used by the Dr. Soetomo General Academic Hospital for more than 20 years for data collection and monitoring of allergy patients.

### 2.4. Outcomes

SCIT is given to patients whose symptoms have not been controlled by symptomatic therapy. According to the Indonesian Pediatric Association, SCIT can be given to patients whose symptoms cannot be controlled with regular allergy medication for at least three months. This includes AR patients who experience wheezing, coughing, and shortness of breath. In AR patients with asthma, spirometry tests were performed. After proper treatment and after their asthma was well-controlled with normal spirometry tests, SCIT was started.

To ensure the safety of these patients, we took an extra procedure for AR who had comorbid asthma; our private allergy clinic performed a PEF test before each SCIT injection. In patients with overt asthmatic symptoms, the injection episode is postponed. Patients with asthma are not advised to take bronchodilator drugs before the scheduled injection. The observation time after injection is 30 minutes. Any sign or symptom assessed as potentially related to the SCIT injection was considered an SE. SE against SCIT is classified into two categories: local SE reactions and systemic SE reactions. Redness, itching, or swelling represent a local reaction at the injection site. Measures of local reactions, such as wheals or deep-itching erythema, were measured and reported on the SCIT administration form given to the patient. We defined significant local reactions as >5 cm in diameter. Parents/caregivers of patients or patients are instructed to record and measure skin reactions that occur at home (late reactions). Local and systemic types of SE are listed on the SCIT administration form at the clinic. Systemic reactions are life-threatening, ranging from mild to very severe anaphylaxis [21]. Systemic reactions are skin symptoms (generalized pruritus, urticaria, flushing, and angioedema), rhinoconjunctivitis, asthma, cardiovascular symptoms, and nonspecific systemic symptoms (headache, cough, vomiting, chest tightness, chest discomfort, etc.). We defined anaphylaxis according to the EAACI [22]. If an SE interferes with the treatment program, it is considered a severe SE. Initial administration of antihistamines before SCIT injection is not used in our clinic. We defined SEs in the SCIT group as unwanted effects that occurred when SCIT-group patients were given an SCIT injection (regardless of the dose), whereas adverse event reactions (AE) in the non-SCIT group we defined as undesirable events resulting from the correct medication in the non-SCIT-group patients that occurred at the same time as the SCIT group received the SCIT injection (Figure 2). Our clinic provides intra-muscular (IM) epinephrine as the first-line treatment of SCIT systemic SE at a 0.01 mg/kg dose. Epinephrine can be repeated every 5 minutes if symptoms have not improved. Second-line treatment is the administration of antihistamines to relieve skin and gastrointestinal symptoms and the administration of beta-agonists to relieve bronchospasm. Corticosteroids are given to prevent biphasic symptoms.

### 2.5. Statistical Analysis

The data was first tested using the one-sample Kolmogorov–Smirnov for normality test. Subsequently, data on patient characteristics were analyzed using the independent t-test or the Mann–Whitney test. In addition, the results of other measurements were analyzed using an independent t-test or the Mann–Whitney test and a dependent t-test or the Wilcoxon test. The data description is mainly based on the mean and standard deviation (SD), or frequency for categorical data. Comparisons between means were made using Student’s t-test. Frequency comparisons were made using a 2 × 2 contingency table, analyzed by the chi-square test. The statistical test results were declared significant if *p* < 0.05. Data analysis used IBM SPSS Statistics software version 23.0 (IBM Corp., Armonk, NY, USA).

## 3. Results

### 3.1. Baseline Characteristics

Figure 1 shows the results of the sample identification procedure. Among all the patients enrolled in the private practices of allergy immunology consultants (*n* = 7356), 83.7% (6126/7356) were diagnosed with AR; among the 6126 patients with newly diagnosed AR, 47.8% (2920) received SCIT de novo. Overall, 1797 SCIT patients and 2313 control patients met the matching requirements; from the existing patient medical records, 1098 SCIT patients were matched with 1098 control patients (non-SCIT). The distribution of children with allergic rhinitis recruited as research subjects based on the geographic area in Indonesia is shown in Figure 3. A total of 2196 allergic rhinitis patients recruited as study subjects came from 92 districts in the Republic of Indonesia.

Table 1 shows stratification according to sex, age, weight/height, disease comorbidities, and disease severity. Of the 1098 patients in the SCIT group, they were predominantly male (63.4%), had the mean age at initial AR diagnosis of 5.5 (SD 3.51) years, had a mean weight at initial AR diagnosis of 12.8 (SD, 2.35) kilograms, and had the mean height at the initial AR diagnosis of 84.7 (SD, 19.83) centimeters. The geographical distribution of the region where they come from is: East Java Region 1 (Surabaya) 43.9%, East Java Region 5 (Bojonegoro) 37.1%, East Java Region 4 (Jember) 4.2%, East Java Region 3 ( Madiun) 2.8%, and East Java Region 2 (Malang) 1.3%, as well as from outside the province (0.4%) and Java (10.1%).

As shown in Table 1, compared with patients in the non-SCIT matched group, patients in the SCIT group experienced a significantly higher burden of comorbid disease overall in the year before SCIT initiation. Whereas asthma and atopic eczema occurred significantly less frequently among patients in the non-SCIT matched group, rates of other upper respiratory diseases such as sinusitis and other respiratory system diseases, such as bronchitis, were significantly higher among patients in the SCIT group. Among the 1098 patients in the SCIT group, there were 696 male patients (63.4%) with a mean age of 5.5 (SD 3.51). Included in this study were those who received SCIT from 2015 to 2020 (see Table 2). Patient characteristics are reported in Table 1. In the year before their initial AR diagnosis, the majority (87.3%) of these AR patients had a comorbid disease burden of asthma, atopic dermatitis, sinusitis, and conjunctivitis of 54%, 9.1%, 1.1%, and 0.3%, respectively. The majority of the severity of the initial AR diagnosis was moderate, with the percentage of mild, moderate, and severe severity being 46.6%, 46.6%, and 6.7%, respectively.

### 3.2. Safety

In Table 2 it can be seen that among 1098 group-SCIT patients who received SCIT injections, there were 284 patients (25.87%) who had SE, and SE that occurred was 699 times or 2.27% of the 30,744 SCIT injections given.

Among 1098 group-SCIT patients who received SCIT injections, there were 196 patients (17.25%) who had experienced local SE, and local SE that occurred were 537 times or 1.75% of the 30,744 SCIT injections given. The numbers of local SE decreased over time, namely at 0–3 months, 4–6 months, 7–9 months, 10–12 months, and 13–18 months they were 200, 157, 118, 38, and 24, respectively. Among 1098 group-SCIT patients who received SCIT injections, there were 92 patients (8.38%) who had experienced systemic SE, and 162 times systemic SE or 0.53% of the 30,744 SCIT injections given. The trend of systemic SE over time consecutively at 0–3 months, 4–6 months, 7–9 months, 10–12 months, and 13–18 months was 69, 42, 9, 24, and 8 times, respectively.

A total of 16 events (100%) of anaphylactic reactions occurred in the first SE. There was one episode of severe anaphylaxis (with symptoms of severe dyspnea, urticaria, and anaphylactic shock), but adrenaline was only needed once to improve it. Most of the SE (468, 70%) occurred in the build-up phase. Subjects in the SCIT group experienced a 0.03% lower incidence of rhinoconjunctivitis than the non-SCIT group, and 0.078% of rhinoconjunctivitis in non-SCIT was caused by infection, while in SCIT it was only 0.03%.

In Table 3, it can be seen that among the 1098 patients in the non-SCIT Group, at the same time as the SCIT group was receiving SCIT injections, none of the patients in the non-SCIT group experienced local AE, but there were 18 patients (1.64%) in the non-SCIT group. SCIT had experienced systemic AE with a total incidence of 46 times (0.15%). Systemic AEs that occurred sequentially (*n*, %) were rhinoconjunctivitis (40, 0.13%), asthma (13, 0.04%), and urticaria (2, 0.01%). The trend of systemic AE over time in a row at 0–3 months, 4–6 months, 7–9 months, 10–12 months, and 13–18 months is up and down, while in a row, the incidence is 9, 15, 15, 1, and 6.

In Figure 4, it can be seen that the occurrence of systemic reactions (systemic AE in the SCIT group and systemic AE in the Non-SCIT group) was significantly different in the 0–3 months, 4–6 months, and 13–18 months periods and was not significantly different in the 7–9-months and 10–12-months periods. Meanwhile, local reactions (local SE in the SCIT group and local AE in the non-SCIT group) were significantly different in all periods.

### 3.3. Efficacy

In Figure 5A, the nasal symptom points of itchy (mean, SD) periods of 0–3 months and 4–6 months in the SCIT group were not different from the non-SCIT group. At 7–9 months, 10–12 months, and 13–18 months, the points were lower for the SCIT group compared to the non-SCIT group. In Figure 5B, the nasal symptom points of runny nose (mean, SD) for 0–3 months in the SCIT group are not different from the non-SCIT group. At 4–6 months, 7–9 months, 10–12 months, and 13–18 months, the points were lower for the SCIT group compared to the non-SCIT group. In Figure 5C, the nasal symptom points of runny nose (mean, SD) for the 0–3-month period in the SCIT group is not different from the non-SCIT group. At 4–6 months, 7–9 months, 10–12 months, and 13–18 months, the points were lower for the SCIT group compared to the non-SCIT group. In Figure 5D, the nasal symptom points of blocked nose (mean, SD) for 0–3 months and 4–6 months in the SCIT group were not different from the non-SCIT group. At 7–9 months, 10–12 months, and 13–18 months, the points were lower for the SCIT group compared to the non-SCIT group. In Figure 5E, the points of eye symptoms (mean, SD) for the 0–3-month period in the SCIT group is not different from the non-SCIT group. At 4–6 months, 7–9 months, 10–12 months, and 13–18 months, the points were lower for the SCIT group compared to the non-SCIT group. In Figure 5F, the points of lung symptoms (mean, SD) for the 0–3-month period in the SCIT group is not different from the non-SCIT group. At 4–6 months, 7–9 months, 10–12 months, and 13–18 months, the points were lower for the SCIT group compared to the non-SCIT group. In Figure 5G, the total points of symptoms (mean, SD) for the periods 0–3 months and 4–6 months in the SCIT group are not different from the non-SCIT group. At 7–9 months, 10–12 months, and 13–18 months, the points were lower for the SCIT group compared to the non-SCIT group.

In Figure 5H and Figure 5I, it can be seen that the mean diameter of the HDM allergen SPT (mm) and the average sp-HDM-IgE level (kU/mL) at the start of therapy in the SCIT group were not different from the non-SCIT group, but at the end of treatment (18 months) in the SCIT group it was lower than in the non-SCIT group.

Table 4 shows the effect of SCIT on medication scores. The mean medication score (MS) of the first trimester (0–3 months) in the SCIT group was 2.6 (SD 0.48), the same as in the non-SCIT group. The decline in MS to 1.7 at 4–6 months to 0.8 at 7–12 months and to 0.4 at 13–18 months occurred in the SCIT group. Meanwhile, for non-SCIT, the MS score decreased to 2.4 at 4–6 months and to 2.2 at 7–12 months, and remains 2.2 at 13–18 months. There was a significant difference in the decline in MS at SCIT from 4–6 months to 13–18 months. The reduction in the MS score on SCIT was 2.3 (SD 0.59), significantly different (*p* = 0.000) from the reduction in the MS score on non-SCIT [0.4 (SD 0.49)].

## 4. Discussion

We found that HDM SCIT could be considered safe in children with AR. In addition, during the maintenance phase, one subcutaneous injection per 3 weeks is sufficient to maintain the efficacy of SCIT. Local SEs in pediatric patients were common in our study. A total of 17.9% of our patients had local SE during SCIT. This contrasts to a study by Yang et al. [23], who reported that 70% of their patients had local SE during SCIT. However, our study is similar to that reported by Yang et al. [23] and Li et al. [24], who found that most local SE occurred with a high-dose injection of HDM extract. The local SE we got was similar to the study results reported by Di Bona et al. [25], which was 17.9% (76.8% of the total SE of 23.3%). This indicates that SCIT in children is well-tolerated. Several reports suggest that local SE during SCIT can occur in up to 93% of patients [26,27,28]. The difference in the incidence of local SE may be due to the types and extracts of allergens used in different studies [29].

We found the incidence of systemic SE in 8.38% of pediatric patients receiving SCIT and 0.53% of the total number of SCIT injections. These results are higher than those reported by the AIT study group (4.94%) [25] and lower than those reported in China (12.09%) [23], as well as from several other studies [30,31,32]. In several studies of Chinese patients, systemic SE occurred in 12.26–18.49% of patients receiving SCIT and 0.72–3.28% of the total number of SCIT injections [30,31,32,33,34]. Consistent with previous studies [33,34], only 2/92 (2.18%) of our patients had a reaction within 30 minutes of injection, and these patients responded well to salvage treatment. The incidence rate in our study was lower than that reported by Yang et al., which was 4.55% [23].

Several studies have shown that the incidence of non-fatal systemic SE in SCIT varies. Urticaria followed by asthma was the most common systemic SE we perceived; this is different from the study results reported by the AIT study group [25], which found that most cases of urticaria followed rhinoconjunctivitis. From studies evaluating SCIT data from 1981 to 1990 [35], systemic SE was found in 5.2% of patients receiving SCIT and 0.06% of the total number of SCIT injections. From studies evaluating SCIT data from 1991 to 2000 [36] in Italy, systemic SE was found in 1.08% of patients receiving SCIT and 0.01% of the total number of SCIT injections. Meanwhile, studies evaluating SCIT data between 1990 and 2001 showed that the rate of unconfirmed systemic SE was 5.4 events per 1 million SCIT injections [37]. Another study by Phillips and colleagues [38] reported systemic SE occurring in 4% of patients receiving SCIT. From studies evaluating SCIT data from 1991 to 2000 in the United States, systemic SE was found in 0.1% of patients receiving SCIT, of which 0.003% was severe anaphylaxis. Evaluation of SCIT data for 2001 and 2002 shows the incidence rate remains the same [39]. In our study, severe anaphylaxis occurred in 0.091% of the 1098 patients in the SCIT group and 0.0033% of the 30,774 SCIT injections.

Studies that included 80% to 85% of allergy practices in the United States from 2008 to 2016 obtained systemic SE in 0.1% of all SCIT injections (we obtained 0.53% systemic SE). With a grading system in four levels of severity (grade 1, mild; grade 2, moderate; grade 3, severe; and grade 4, very severe), from studies evaluating SCIT data from 2012 to 2016, systemic SE was found in 0.087% of patients who received SCIT (8.7 systemic SE per 10,000), with a breakdown of 0.056% grade 1, 0.027% grade 2, 0.0035% grade 3, and 0.000625% grade 4 [40, 41]. From 2008 to 2016, non-fatal systemic SE did not increase; specifically, grade 1 (mild) systemic SE showed a downward trend. In general, the systemic SE level of SCIT has remained stable, although an increase in mortality has been reported in recent years [40,41].

These data indicate that SCIT is safe for children. To ensure the safety of these patients, we took an extra procedure for AR who had comorbid asthma by having a PEF test before each injection. We need to strictly ensure patient safety because uncontrolled asthma contributes to fatal systemic SE. In the 1990 and 2001 surveys that studied fatal systemic SE, it was found that the incidence of fatal systemic SE was one in every 2.5 million injection visits or about 3.4 fatal reactions per year. Most patients with fatal systemic SE had poor asthma control before the visit [42]. The weakness in young children is the lack of accurate PEF data. Patients will delay their injections if the PEF value does not reach 80% of the predicted value to reduce the incidence of systemic SE in children.

Taking the extra step we have taken in our patients, we found that the incidence of systemic SE in our patients after SCIT (by 8.38% of patients receiving SCIT, 0.53% of the total number of SCIT injections) was relatively the same as the incidence in adults (5.68–10.98% of patients receiving SCIT, 0.31–1.47% of the total number of SCIT injections) as reported by several studies [32,43,44]. Given that most systemic SEs occur when high-dose HDM allergen injections are given, the maximum tolerable dose in this population needs further investigation to strike a balance between efficacy and safety. The dose of HDM allergen for SCIT will be reduced appropriately after systemic SEs develop if SCIT continues. We usually reduce the dose to a previously tolerable dose or even a lower dose if the reaction is severe [45]. Our study found that anaphylactic episodes occurred in 16 patients (1.46%) and in 15 patients (1.37%) who had first and second episodes. We only found one severe episode and it was resolved with the use of adrenaline only once. The AIT study group [25] reported that 29 patients (1.3%) experienced an episode of anaphylaxis, with two severe episodes that resolved with one-time use of adrenaline. Our results show that most systemic SEs occur in the build-up phase, like in other studies.

The HDM allergen extract for SCIT that we used was also proven to improve AR as indicated by the parameters points of nasal, eye, and lung symptoms, and total points of symptoms at 0–3 months, 4–6 months, 7–9 months, 10–12 months, and 13–18 months, as well as a decrease in the diameter of the skin prick test and the level of spHDM IgE in kU/mL, which was better in the SCIT group compared to the non-SCIT group, as the results of other previous studies [1,2,3,4].

### Strength and Limitation

The advantage of our study is that, apart from evaluating the incidence of adverse events (AE) in the SCIT group after SCIT injection, we also assess the incidence of adverse events (AE) in the non-SCIT group as a control in the same week. Systemic reactions (systemic SE in the SCIT group and systemic AE in the non-SCIT group) were significantly different in the periods of 0–3 months, 4–6 months, and 13–18 months and were not significantly different in the periods of 7–9 months and 10–12 months. Meanwhile, local reactions (SE local in the SCIT group and local AE in the non-SCIT group) were significantly different in all periods. Thus, our study is not only a descriptive study of the safety profile but also confirms that the occurrence of systemic and local SE in SCIT in this study is a true occurrence, not just by chance, and can occur in the non-SCIT group as well. Several limitations should be mentioned in this study. However, we have attempted to match patients with potentially confounding variables but may have been unable to control for other important characteristics, such as patient adherence to pharmacological treatment and allergen avoidance. Allergen avoidance in the allergy guidelines of the Indonesian Pediatric Society is recommended as the first step in allergy treatment. Still, there is no guarantee that all families will comply, so we cannot determine whether SCIT- and non-SCIT-matched patients are equally likely to comply with instructions on allergen avoidance. In addition, we also do not have information regarding the implementation of the avoidance measures that we recommend to parents.

## 5. Conclusions

Our study concluded that in pediatric patients with AR who received HDM SCIT for 18 months with high adherence, some experienced significant local SEs and systemic SEs because of SCIT, but this did not interfere with the course of AR treatment or the effectiveness of SCIT.

## Figures and Tables

**Figure 1 cells-11-01584-f001:**
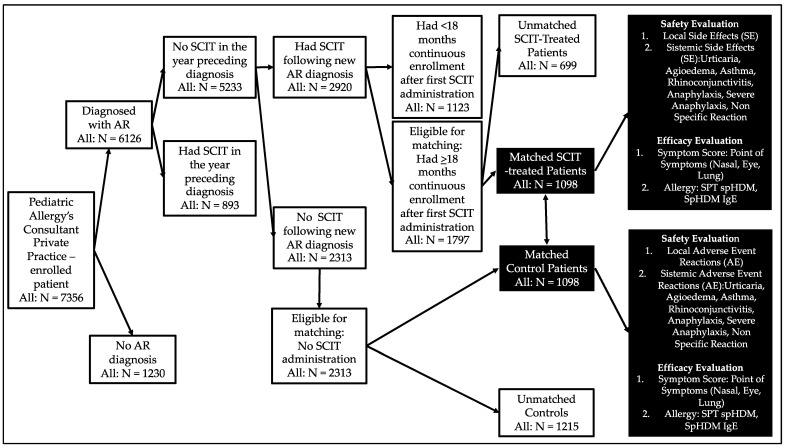
Identify suitable samples. Note: The eight matched variables were age, sex, weight, height, family history of allergies, symptoms, and treatment scores at SCIT initiation and comorbid atopic conditions (asthma, conjunctivitis, and atopic dermatitis) during the year before SCIT initiation.

**Figure 2 cells-11-01584-f002:**
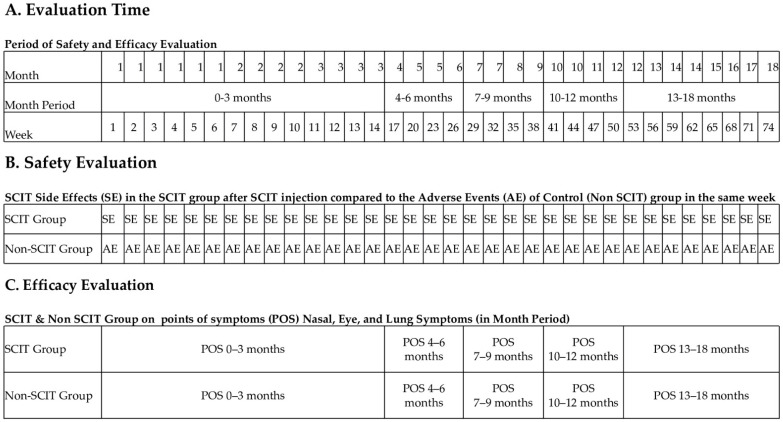
Evaluation of side effects (SE) in the SCIT group after SCIT injection compared to the adverse events (AE) of the control (non-SCIT) group in the same week that accumulated in 0–3 months, 4–6 months, 7–9 months, 10–12 months, and 13–18 months (**A**,**B**), as well as evaluation of points of symptoms (POS) in the SCIT and non-SCIT groups in the same month period (**C**).

**Figure 3 cells-11-01584-f003:**
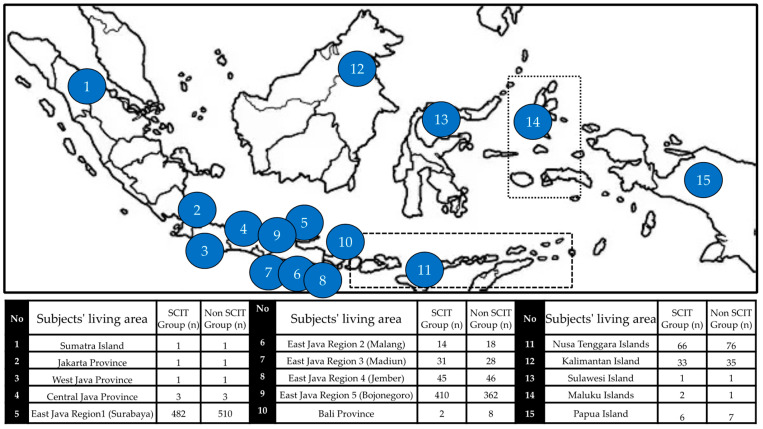
Distribution of 2196 children with allergic rhinitis were recruited as the subjects of this study by geographic area in Indonesia (island, province, or region).

**Figure 4 cells-11-01584-f004:**
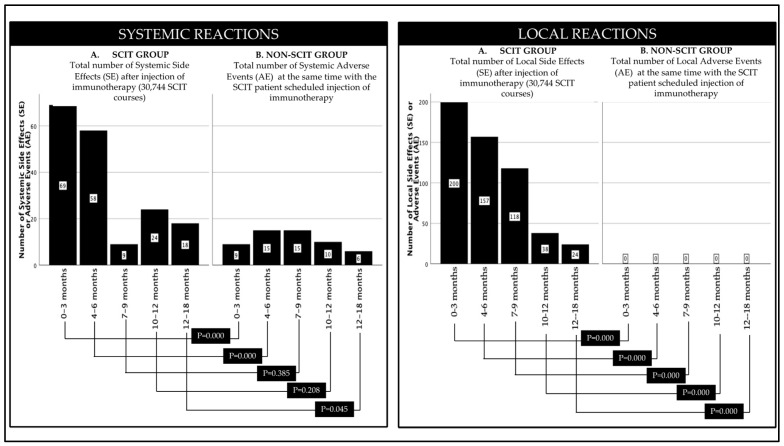
Number of subjects experiencing systemic reactions and local reactions. Systemic reactions after SCIT-group patients received SCIT injections were compared with the incidence of adverse reactions experienced by non-SCIT-group patients at the same time because of drug or allergic disease, as well as local reactions.

**Figure 5 cells-11-01584-f005:**
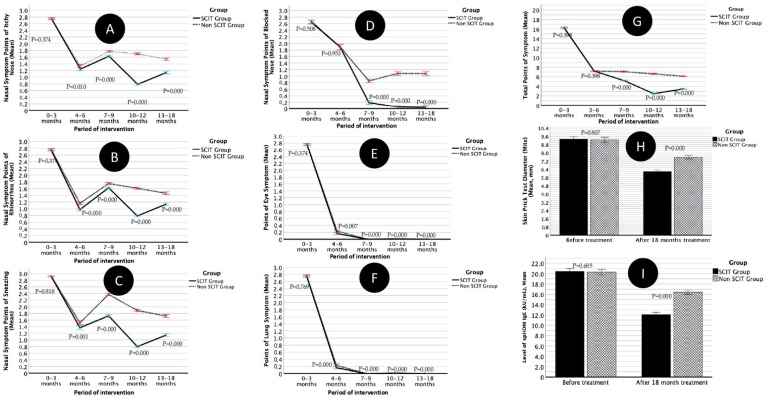
Differences in points of nasal (**A**–**D**), eye (**E**), and lung symptoms (**F**), and total points of symptoms (**G**) at 0–3 months, 4–6 months, 7–9 months, 10–12 months, and 13–18 months, as well as differences in skin prick test diameter in mm (**H**) and IgE spHDM levels in kU/mL (**I**) between the SCIT group and the non-SCIT group.

**Table 1 cells-11-01584-t001:** Baseline Characteristics of Study Participants.

Characteristics	SCIT Group (*n* = 1.098)		Control Group (*n* = 1.098)		*p*
Age (years), mean (SD)	5.5	(3.51)	5.4	(3.32)	0.465
Sex, *n* (%)					
Male	696	(63.4)	704	(64.1)	0.722
Female	402	(36.6)	394	(35.9)	
AR-associated conditions, *n* (%)					
Asthma, *n* (%)	593	(54.0)	396	(36.1)	0.000
Bronchitis, *n* (%)	431	(39.3)	700	(63.8)	0.000
Atopic dermatitis, *n* (%)	100	(9.1)	85	(7.7)	0.249
Sinusitis, *n* (%)	12	(1.1)	106	(9.7)	0.000
Conjunctivitis, *n* (%)	3	(0.3)	0	(0.0)	0.083
GI Problem, *n* (%)	7	(0.6)	3	(0.3)	0.205
Urticaria, *n* (%)	14	(1.3)	20	(1.8)	0.300
Nutrition Status					
BW (kgs), mean (SD)	12.8	(2.35)	12.7	(2.33)	0.761
BH (cm), mean (SD)	84.7	(19.84)	84.2	(19.24)	0.530
% BW/Age	82.3	(3.29)	82.3	(3.26)	0.938
% BH/Age	77.9	(4.11)	77.9	(4.07)	0.938
Geographic region, *n* (%)					
East Java Region 5 (Bojonegoro)	410	(37.3)	362	(32.9)	0.325
East Java Region 4 (Jember)	45	(4.1)	46	(4.2)	
East Java Region 3 (Madiun)	31	(2.8)	28	(2.5)	
East Java Region 2 (Malang)	14	(1.3)	18	(1.5)	
East Java Region 1 (Surabaya)	482	(43.9)	510	(46.4)	
Outer Geographic Region	115	(10.5)	138	(12.5)	
Symptom score (SS) before treatment, mean (SD)	2.7	(0.48)	2.7	(0.45)	0.398
Skin prick test diameter (mite) before treatment (mm), mean (SD)	9.3	(4.17)	9.3	(4.15)	0.807
spHDM IgE (kU/mL), mean (SD) before treatment (checked randomly by 10% of the total sample)	20.5	(8.75)	20.3	(8.66)	0.685

**Table 2 cells-11-01584-t002:** Side effects (SE) after injection of immunotherapy (30,744 SCIT courses).

Side Effects (SE)	Total Number of Side Effects (SE) after Injection of Immunotherapy (30,744 SCIT Courses)							Total Patients with SE (N SCIT Group = 1098)		Mean of SE/Subject Who Experience SE
	0–3 Months	4–6 Months	7–9 Months	10–12 Months	13–18 Months	Total 0–18 Months				
						*n*	%	*n*	%	
1. Local	195	156	117	38	24	530	1.72	195	17.76	2.72
2. Urticaria	27	3	0	24	0	54	0.18	27	2.46	2.00
3. Angioedema	1	0	1	0	0	2	0.01	2	0.18	1.00
4. Asthma	9	9	7	0	7	32	0.10	16	1.46	2.00
5. Rhinoconjungtivitis	11	19	0	0	0	30	0.10	19	1.73	1.58
6. Nonspecific	0	11	0	0	11	22	0.07	11	1.00	2.00
7. Anaphylaxis	15	15	0	0	0	30	0.10	15	1.37	2.00
8. Severe anaphylaxis	1	0	0	0	0	1	0.00	1	0.09	1.00
9. Local and systemic	5	1	1	0	0	7	0.02	1	0.09	7.00
10. Total local	200	157	118	38	24	537	1.75	196	17.85	2.74
11. Total systemic	69	42	9	24	18	162	0.53	92	8.38	1.76
Total Side Effects (SE)	269	199	127	62	42	699	2.27	284	25.87	2.46

**Table 3 cells-11-01584-t003:** Adverse events (AE) in non-SCIT patients at the same time with the SCIT-patient scheduled injections of immunotherapy.

Adverse Events (AE)	Total Number of Adverse Events (AE) of Non-SCIT Patients at the Same Time with the SCIT-Patient Scheduled Injection of Immunotherapy							Total Patients with AE (N Non-SCIT Group = 1098)		Mean of AE/Subject Who Experience AE
	0–3 Months	4–6 Months	7–9 Months	10–12 Months	13–18 Months	Total 0–18 Months				
						*n*	%	*n*	%	
1. Local	0	0	0	0	0	0	0	0	0	
2. Urticaria	1	0	0	0	1	2	0.01	1	0.09	2
3. Angioedema	0	0	0	0	0	0	0	0	0	
4. Asthma	3	7	2	1	0	13	0.04	4	0.36	3.25
5. Rhinoconjungtivitis	5	8	13	9	5	40	0.13	13	1.18	3.08
6. Nonspecific	0	0	0	0	0	0	0	0	0	
7. Anaphylaxis	0	0	0	0	0	0	0	0	0	
8. Severe anaphylaxis	0	0	0	0	0	0	0	0	0	
9. Local and systemic	0	0	0	0	0	0	0	0	0	
10. Total local	0	0	0	0	0	0	0	0	0	
11. Total systemic	9	15	15	1	6	46	0.15	18	1.64	2.56
Total Adverse Events (AE)	9	15	15	1	6	46	0.15	18	1.64	2.56

**Table 4 cells-11-01584-t004:** Effect of SCIT on Medication Scores.

Medication Score (MS), Mean (SD)	SCIT Group (*n* = 1.098)		Control Group (*n* = 1.098)		*p*
0–3 months	2.6	(0.48)	2.6	(0.48)	0.860
4–6 months	1.7	(0.53)	2.4	(0.48)	0.000
7–12 months	0.8	(0.61)	2.2	(0.69)	0.000
13–18 months	0.4	(0.59)	2.2	(0.69)	0.000
Difference Before-After	2.3	(0.59)	0.4	(0.49)	0.000

## Data Availability

The authors confirm that the data supporting the findings of this study are available within the article.

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
