# Peer review of "Safety Profile and Issues of Subcutaneous Immunotherapy in the Treatment of Children with Allergic Rhinitis"

_cells, 2022, doi:10.3390/cells11091584_

Round 1
Reviewer 1 Report
The authors investigated the safety profiles of house dust mites subcutaneous immunotherapy among 1098 allergic rhinitis children in a retrospective cohort study. They found there were 284 patients (25.87%) who had side effects, and side effects that occurred was 699 times or 2.27% of the 30,744 subcutaneous immunotherapy injections given. They concluded a few incidences of systemic reactions associated with house dust mite subcutaneous immunotherapy and although local reactions were common, it did not interfere with the effectiveness of subcutaneous immunotherapy. This study presents the safety and efficacy data of HDM SCIT in a relatively large-size population. However, there are several questions to be addressed.
- Why did the authors set a control group (no SCIT group) in this study? For safety issue, it’s easy to figure out SCIT-related AEs in most cases and the AEs in the control groups is not necessary to demonstrate the safety profile of SCIT. However, it may be interesting to know if SCIT reduced respiratory infections in this population. For efficacy issue, how did the authors assure the patients in the two groups followed the same medication treatment protocol in a retrospective cohort study?
- The authors need to explain the method to match the two groups, by gender, age or other variables? Some demographic characteristics were not balanced in the two groups, which might have impact on efficacy and safety of SCIT.
- The medication scores were not recorded in the study, the international guideline suggested to evaluate the efficacy of SCIT by combined symptom and medication score in AR patients.
- It should be noted some patients would dropped out of SCIT within 18 months, which might weaken their conclusion.
- The quality of allergen extracts has a significant impact on SCIT safety and efficacy, and the WAO recommend to provide product-based evidence of immunotherapy. I suggest the authors focus on the risk factors related to SCIT-related AEs and efficacy, which will be more interesting to readers, but instead of the direct comparison of safety data with different allergen products in other areas.
Reviewer 2 Report
Peer review of the article entitled: “Safety profile and issues of subcutaneous immunotherapy in the treatment for children with allergic rhinitis”
General Comments:
The main objective of this article was to evaluate safety profiles among allergic rhinitis children receiving house dust mites subcutaneous immunotherapy. A retrospective cohort study had been done from 2015 until 2020 to investigate any side effects of subcutaneous immunotherapy among rhinitis children due to house dust mite allergy. Among 1098 patients who received house dust mite subcutaneous immunotherapy injections, there were 284 patients (25.87%) who had side effects, and side effects that occurred was 699 times or 2.27% of the 30,744 subcutaneous immunotherapy injections given. This study demonstrates a few incidences of systemic reactions associated with house dust mite subcutaneous immunotherapy. Local reactions are common; however it does not interfere with the effectiveness of subcutaneous immunotherapy.
Specific Comments: The Abstract should be improved to make it more comprehensible. Data should be presented by number of patients with reactions (or % of total patients) and by % of injections given. Reactions should also be presented as local or systemic reactions. The size and severity of the reactions should also be graded according to the EAACI guidelines on allergen immunotherapy.
The text contains several sections in which the English language should be revised. Some areas are not clear and the revision by a native English speaker is recommended. Fo example:
Another study also stated that the most common HDM in Indonesia Dermatophagoides pteronyssinus is, which can be found in various places such as beds, floors, and sofas, while Dermatophagoides farinae is most often found on sofas. Bromia tropicalis is the least expensive compared to Dermatophagoides pteronyssinus and Dermatophagoides farinae [20].
Mistakes are also present in other areas of the document. Please revise.
Are the extract native allergen preparations? They are not Allergoids.
Any indication on the production process ? Major allergen content ?
Final presentation of the vaccine?
These data are important in order to compare the results with other studies.
Round 2
Reviewer 1 Report
The authors need to clarify the criteria of SCIT-related AE in the draft.